# Hygiene Behaviors and SARS-CoV-2-Preventive Behaviors in the Face of the COVID-19 Pandemic: Self-Reported Compliance and Associations with Fear, SARS-CoV-2 Risk, and Mental Health in a General Population vs. a Psychosomatic Patients Sample in Germany

**Sonia Lippke** [1,*] , **Franziska M. Keller** [1] , **Christina Derksen** [1] , **Lukas Kötting** [1] **and Alina Dahmen** [1,2,3]

1. Department of Psychology & Methods, Jacobs University Bremen, 28759 Bremen, Germany; f.keller@jacobs-university.de (F.M.K.); c.derksen@jacobs-university.de (C.D.); l.koetting@jacobs-university.de (L.K.); adahmen@dbkg.de or Alina.Dahmen@Klinikum.Wolfsburg.de (A.D.)
2. Becker Klinikgruppe, 50968 Köln, Germany
3. Klinikum Wolfsburg, 38440 Wolfsburg, Germany
*   Correspondence: s.lippke@jacobs-university.de; Tel.: +49-421-2004730

**Abstract:** *Background*: During the COVID-19 pandemic, hygiene behaviors such as keeping distance, avoiding masses, wearing face masks, and complying with hand hygiene recommendations became imperative. The current study aims to determine factors interrelating with hygiene behaviors. *Methods*: A total of 4049 individuals (1305 male, 2709 female, aged 18–80 years) were recruited from rehabilitation clinics or freely on the internet. They were surveyed via online questionnaires between May 2020 and August 2021. Sociodemographics, hygiene behaviors, and fear of COVID-19 infection were assessed. *Results*: Overall prevalence for hygiene behaviors was: keeping a distance—88.1%; avoiding mass gatherings—88.0%; wearing face masks—96.9%; and hand hygiene—81.6%, with 70% of the study participants complying with all four researched behaviors. Hygiene behaviors were significantly related to fear in a linear and quadratic fashion. *Conclusion*: Patients are more compliant according to their self-reported responses than the general population. To improve hygiene behavior, hand hygiene in particular provides options for improvements. A medium level of fear seems to be more functional than too-elevated fear. Behavioral interventions and targeted communication aiming at improving different behaviors in orchestration can help individuals to protect their health and to remain healthy. Accordingly, communication is required to ensure high hygiene standards and patient safety, and to prevent adverse effects.

**Keywords:** COVID-19; medical rehabilitation; psychosomatic patients; general public; infection; physical health; psychological health

## 1. Introduction

The prevention of communicable diseases and infections is key for population health and patient safety, and protecting the health of susceptible populations [1]. This becomes especially clear during the COVID-19 pandemic with the risk of transmissions from human to human [2]. *Hygiene behaviors* are not new, but during the COVID-19 pandemic, the population was required to perform behaviors such as:

*   Keeping a distance of 1.5 m between humans;
*   Avoiding mass gatherings;
*   Wearing face masks;
*   Complying with hand hygiene requirements [3].

The first three behaviors have been required by law in Germany at most times during the pandemic. In contrast, hand hygiene is recommended and enforced by policies for

institutions such as hospitals, senior homes, and care homes for susceptible populations [4], but not by regulation on the population level, according to the German Law of Infection prevention Law, Section 28a and 28b [4].

The COVID-19 pandemic led governments and responsible parties, as well as the press, to communicate these behaviors in many countries, including Germany [5,6]. Accordingly, it was often found that the level of *fear* (negative emotion relating to anxiety, worry, or concern of own risk) of infection with the SARS-CoV-2 virus was interrelated with the prevalence of frequent hand hygiene [7,8] but not with physical distancing, leaving home for food or to exercise [8]. Additionally, the fear of becoming seriously ill with COVID-19 was correlated with behavior change [9]. Vulnerability, perceived risk, and fear were all found to be significantly interrelated with preventive behaviors during the COVID-19 pandemic [10], but the results are not always consistent [3]. It has been found that fear can be buffered by hand hygiene behavior [11] and thereby work functionally [9]. However, whether this is the case with fear and other hygiene behaviors, too, has not been systematically tested so far relating to the COVID-19 pandemic in Germany.

Although we can learn from the past with campaigns preventing the spread of, e.g., hospital transmissions and hospital-acquired infections (HAIs [12]), the COVID-19 pandemic affects the whole of society on a much larger scale. Thus, hygiene has also come into focus among people who previously had no awareness or knowledge of infection prevention: the COVID-19 pandemic has changed many areas of life, and directly or indirectly impacted everyone worldwide. It involved emotions such as fear [13], which were intentionally addressed by the media and social environment [14]. Especially in vulnerable groups, the COVID-19 pandemic increased existing fear [15,16]. It is well known from the findings of the influenza A pandemic that increased media coverage leads to an increase in anxiety [17].

### 1.1. COVID-19 Related Fear and Behavior

Fear, as an adaptive response to a perceived threat or danger, has been reported to be the first emotional response to the COVID-19 pandemic. Thus, fear of infection and possible consequences can either be irrational or rational and commonly concern getting infected, transmitting the disease, losing relatives or possible negative health outcomes (i.e., post-COVID) [18,19]. Although people with *risk factors for COVID-19* have received considerable early attention [2,20], less attention has been paid to how other susceptible populations, i.e., individuals with previous mental conditions, cope with the fear of a COVID-19 infection [21]. Although increasingly systematic research is now available on risk factors causing a severe course of COVID-19 (such as cardiovascular disease, diabetes, respiratory illnesses, liver disease, kidney disease or cancer) [22], and the knowledge that *people with health limitations* have more barriers to perform hygiene behavior [23], rather little is known about the differences between the general population and patients in, e.g., medical rehabilitation. What is known from before is the following: Patients with (pulmonary) comorbidities, i.e., coronavirus risk factors, report significantly higher fear levels than those without comorbidities [20].

Over the course of the pandemic, several researchers have examined the link between the fear of infection, enforced isolation, and *mental health symptoms*. It was generally found that uncertainty led to higher levels of anxiety and depression [24,25]. Especially in vulnerable populations who were susceptible or already suffering from mental health problems, social isolation and loneliness were responsible for substantially higher rates of mental health problems [26]. Hence, patients with psychological disorders such as the diagnosis of depression and anxiety disorders require special attention. Potential risks may be particularly relevant for psychosomatic patients due to their history of mental health problems and associated risk factors leading to the rehabilitation stay.

The relationship between fear of infection, behavior and its underlying cognitive and emotional processing has been mostly examined in professional groups working in the health sector [27]. Studies revealed a *dose-dependent relationship* between stress levels and

change in behaviors due to the pandemic with an inverse U-shaped trend [27], which has been termed *Yerkes-Dodson Law* [28]. Adding to the U-shaped association, the dynamic course of epidemics and in this case the COVID-19 pandemic needs to be considered. Usually, compliant hygiene behavior does not last over the course of a pandemic [29]. Therefore, that behavior change needs to be addressed in a dynamic fashion.

*1.2. Relationship between Different Health Behaviors: Theoretical Explanation and Model*

A possible theoretical foundation for the complex relationship between multiple hygiene behaviors is the *Compensatory Carry-Over Action Model* (CCAM [30]). The CCAM models different single activities—such as hand hygiene and face mask wearing—and how they may change as a result of one another. Such lifestyle activities are assumed to be formed by higher-level goals (e.g., striving to stay/remain healthy and uninfected), which can drive activity volitionally or unconsciously, and are rather unspecific. They become specific because of behavior that is subjectively seen as leading to this goal. Each behavior must be intended, pursued, and controlled. Specific resources ensure that individuals have the chance to translate their intentions into activity and that they resist distractors.

*Compensation and transfer* (also called carry-over) operate between the different behaviors. If people devote all of their energy to one domain, they might believe that no resources remain for the other activity. In contrast, the other activity (such as mask wearing) might not be as important anymore when one behavior is displayed (e.g., avoiding mass gatherings). This reflects compensatory thinking and actions. However, as there might also be other instances to get in contact with other individuals a full compensation might not be possible. It would be safer from a hygiene perspective to perform all behaviors adequately. The psychological mechanisms would be the transfer of existing resources (such as the self-efficacy belief) to the other behavior. Both transfer and compensation are described by the CCAM.

Different theories have been used to investigate hygiene behaviors [31,32] but no study with the CCAM could be found so far modelling hygiene behaviors. Although the CCAM does not explicitly incorporate fear, it has been found that fear and health (as a higher-level goal) are interrelated as more fears interrelate with health in patient samples [33]. In another study, behavior change in face of fear of COVID-19 was related to psychological health but not physical health, indicating the role of mental health in COVID-19 related behavior change [9].

*1.3. Research Questions*

Based on these previous findings regarding hygiene behavior during the COVID-19 pandemic and consequences on health and well-being of the general population and psychosomatic rehabilitation patients, this study aims to offer new insights into the dynamic process of health behavior and fear by applying the CCAM. More specifically, the study aims to answer the following research questions:

*Research Question 1: What is the prevalence of hygiene behaviors, overall and in the two samples comparing general population vs. patients with mental health problems?*

*Research Question 2: To what extent are hygiene behaviors (avoiding masses, physical distancing, hand hygiene, and face mask use as well as the aggregated behaviors) correlated with each other overall and in the two samples?*

*Research Question 3: To what extent are hygiene behaviors (avoiding masses, physical distancing, hand hygiene, and face mask use as well as the aggregated behaviors) correlated with overall fear in the two samples?*

The current study aims to test all three research questions in individuals from the general population and from a sample of patients with mental health problems. The procedure will be outlined in the following.

## 2. Materials and Methods

The study was conducted as part of the two projects "Anhand-COVID19—Offer to achieve treatment and rehabilitation goals in compliance with hygiene and social-distancing rules" (ClinicalTrials.gov Identifier: NCT04453475) and "TeamBaby—Safe, digitally supported communication in obstetrics and gynecology" (ClinicalTrials.gov Identifier: NCT03855735). Both studies with their relating data collection procedures will be described below.

### 2.1. First Sample: Recruitment and Procedure of the General Population

Data were collected anonymously through a nationwide recruitment campaign, press releases, social media posts, and the study home page of the TeamBaby project. For data collection purposes, the software tool Unipark was used. All participants were informed about the purpose of the survey beforehand and provided online informed consent. Participants were not offered any form of compensation for participation.

Data collection from the general population took place between May 2020 and August 2021. The time to complete the survey took, on average, 14.46 min (SD 9.47). Ethical approval for the online survey for the general population was given by the Ethics Committee at Jacobs University Bremen on 17 September 2019. The sample characteristics of the general population are reported in Table 1.

**Table 1.** Characteristics of the general population (first sample) and the psychosomatic rehabilitation patients (second sample).

| | General Population (First Sample) N = 1894 | Patients (Second Sample) N = 2155 | Test Statistic |
|---|---|---|---|
| Gender | | | $Chi^2$ (4036,2) = 23.707 ** |
| *Male* | 563 (29.7%) | 742 (34.6%) | |
| *Female* | 1312 (69.3%) | 1397 (65.2%) | |
| *Divers/other* | 19 (1.0%) | 3 (0.1%) | |
| Age [0] | | | $Chi^2$ (4042,4) = 529.405 ** |
| *<30 years* | 441 (23.3%) | 89 (4.1%) | |
| *30–39 years* | 426 (22.5%) | 271 (12.6%) | |
| *40–49 years* | 374 (19.7%) | 498 (23.2%) | |
| *50–59 years* | 393 (20.7%) | 984 (45.8%) | |
| *60+ years* | 260 (13.7%) | 306 (14.2%) | |
| Education (years schooling) | | | $Chi^2$ (4014,3) = 213.344 ** |
| *≤9 years* | 189 (10.0%) | 66 (3.1%) | |
| *10 years* | 405 (21.4%) | 482 (22.7%) | |
| *11 years* | 527 (27.8%) | 979 (46.2%) | |
| *≥12 years* | 773 (40.8%) | 593 (28.0%) | |
| Working | 993 (72.6%) | 1571 (73.3%) | $Chi^2$ (3510,1) = 0.242 |
| Living with a partner/spouse | 1148 (60.0%) | 1328 (61.6%) | $Chi^2$ (4049,1) = 0.434 |
| Living with a child/children | 666 (35.2%) | 741 (34.4%) | $Chi^2$(4049,1) = 0.269 |
| COVID-19 risk factors | 424 (25.3%) | 878 (43.9%) | $Chi^2$ (3680,1) = 138.520 ** |
| Disabled | 128 (7.2%) | 294 (13.8%) | $Chi^2$ (3898,1) = 43.967 ** |
| Disability | 294 (13.8%) | 128 (7.2%) | $Chi^2$ (3898,1) = 43.967 ** |
| Previous COVID-19 infection | 148 (9.9%) | 56 (3.0%) | $Chi^2$ (3349,1) = 68.935 ** |
| Fear of being infected with the coronavirus | 2.27 (1.143) | 2.79 (1.036) | F (3932,1) = 223.079 ** |
| Fear of getting seriously ill with COVID-19 | 2.07 (1.130) | 2.68 (1.108) | F (3926,1) = 292.766 ** |
| Fear of infecting relatives/roommates or friends with COVID-19 | 2.51 (1.249) | 2.88 (1.144) | F (4047,1) = 98.245 ** |
| Fear (Index) | 2.29 (1.064) | 2.79 (0.967) | F (4047,1) = 246.683 ** |
| Change in physical health | 5.23 (1.991) | 4.89 (1.848) | F (3964,1) = 31.863 ** |
| Change psychological health | 4.17 (2.374) | 4.25 (1.981) | F (3865,1) = 1.551 |

[0] Age range 18–80 years; Mean (standard deviation in brackets), ** $p < 0.001$.

Overall, 1894 individuals from the general population were recruited. The majority was female, below 40 years of age and with 12+ years of schooling (Table 1).

*2.2. Second Sample: Recruitment and Procedure of Psychosomatic Rehabilitation Patients*

The second group of participants were recruited through four psychosomatic clinics from the Dr Becker clinic group and attended regular treatment at the recruiting clinics, consisting of psychological and physical interventions (i.e., individual and group psychotherapy, physiotherapy, or occupational therapy) as part of the incoming process for their medical rehabilitation stay. Participation in this study was possible from six weeks before until the first day of the medical rehabilitation treatment. Patients assigned to one of the clinics were informed about the study in writing on the hospital group's original online portal. Therefore, only patients who had access to this digital portal via smartphone, tablet, or computer before the start of rehabilitation were included.

Participation was only possible after the patients had read the participation information and had given their informed consent in writing; data were pseudonymized. Rehabilitation patients were not offered any form of compensation for their participation in the online study. The online survey at the psychosomatic clinics was administered between July 2020 and August 2021. Time to complete the survey took, on average, 22.11 min (SD 15.03). Ethical approval for the online survey concerning psychosomatic rehabilitation patients was given by the Ethics Committee at Jacobs University Bremen on 25 June 2020.

Overall, 2155 individuals from the patient population were recruited. The majority was female, above 50 years of age and with 11+ years of schooling (Table 1).

There were significant differences between the general population sample and the patient sample in 12 out of 16 variables tested on a bivariate level (Table 1).

*2.3. Instruments*

If not differently stated in the following, all instruments were used before in a previous study published in German [34] and revealed adequate measurement qualities.

2.3.1. Hygiene Behaviors

Self-reported hygiene behavior in terms of *hand hygiene* was measured with the item "Do you wash or disinfect your hands before and after every purchase, touching door handles outside your own home, taking public transport, etc.?" The possible response options were (1) "No, I do not intend to.", (2) "No, but I've thought about it.", (3) "No, but I've decided to do it.", (4) "Yes, but it's hard for me.", and (5) "Yes, and it's easy for me.". Answers were dichotomized as "not performing the hygiene behavior/non-compliant" (1 to 3) or "performing the hygiene behavior/compliant" (4 and 5).

Hygiene behavior in terms of *wearing a face mask* was measured with the item "Do you wear a mouth and nose protector every time you go shopping, visit hospitals and use public transportation, etc.?" The possible response options were the same as for hand hygiene. Answers were dichotomized as "not performing the hygiene behavior/non-compliant" (1 to 3) or "performing the hygiene behavior/compliant" (4 and 5).

Hygiene behavior in terms of *avoiding large masses* was asked to indicate the most appropriate response to the statement "I stay away from crowded places or mass gatherings.", with the options (1) "Do not agree at all", (2) "Do rather not agree", (3) "Agree to some extent", or (4) "Agree fully". Answers were dichotomized as "not performing the hygiene behavior/non-compliant" (1 or 2) or "performing the hygiene behavior/compliant" (3 and 4).

Hygiene behavior in terms of *keeping a physical distance of 1.5 m from other individuals* was asked to indicate the most appropriate response to the statement "I keep distance (at least 1.5 m) between me and other people.", with the options (1) "Do not agree at all", (2) "Do rather not agree", (3) "Agree to some extent", or (4) "Agree fully". Answers were dichotomized as "not performing the hygiene behavior/non-compliant" (1 or 2) or "performing the hygiene behavior/compliant" (3 and 4).

All four behaviors were aggregated in terms of a means score which revealed good internal reliability with Cronbach's Alpha = 0.756. All four behaviors were also aggregated to determine how many behaviors study participants are complying with. This variable could vary from 0 (none of the hygiene behaviors performed according to self-report) to 4 (all of the hygiene behaviors complied as indicated by self-report).

### 2.3.2. Fears of a SARS-CoV-2 Infection and COVID-19

Fears relating to the infection with SARS-CoV-2 and possible consequences of COVID-19, such as fears of being infected, fears of getting seriously ill with COVID-19, and fears of infecting close ones, such as relatives, roommates or friends, were measured by three items. All three items were assessed on a five-point Likert scale from (1) "Never", (2) "Rarely", (3) "Sometimes", (4) "Often", and (5) "Always". The items worded "How often do you fear being infected with the coronavirus?", "How often do you fear becoming seriously ill with COVID-19" and "How often do you fear of infecting relatives/roommates or friends with COVID-19". All three fear items were aggregated in terms of a means score, which revealed good internal reliability with Cronbach's Alpha = 0.862.

### 2.3.3. Sociodemographic Characteristics

Additional data on sociodemographic information included participants' age, sex, and educational status. Age was categorized into five groups: ≤29 years, 30–39 years, 40–49 years, 50–59 years, and ≥60 years. Sex was categorized into three groups: male, female, and diverse. The highest obtained educational status was categorized into four groups: 10 or 11 years of schooling, 12 or more years of schooling, vocational training, and university degree.

In addition, employment status was assessed by the item "Are you currently employed? Which one applies best to your status?" which has been adapted from the German-Socio-Economic-Panel (SOEP [35]). Those reporting "Employed full-time", "Employed part-time" or "Completing in-service training/apprenticeship/in-service retraining" were considered as working and all others as not working.

### *2.4. Statistical Analyses*

For all analyses, SPSS Version 28 was used (IBM Corp., Armonk, NY, USA). Bivariate and multivariate statistical analyses were performed. *Research question 1* was tested with frequency analyses (Chi$^2$). *Research question 2 and 3* were tested with correlations analyses (Spearman's rho because hand hygiene and face mask wearing were measured in an ordinal fashion only).

*Research question 3* regarding the trends of the interrelations was analyzed with curve estimations testing for not only the linear trend but whether furthermore quadratic and cubic trends would explain additional variance. In addition, a MANOVA was computed with the different fears as dependent variables and self-reported compliance behavior as well as group (patients vs. general population).

### 3. Results

In this section, the three research questions are tested with the data combined from both samples and with different analyses.

### *3.1. Research Question 1: What Is the Prevalence of the Behaviors, Overall and in the Two Samples Comparing General Population vs. Patients with Mental Health Problems?*

According to overall self-reported hand hygiene and face mask wearing, 81.6% and 96.6% of the study participants were performing the recommended hygiene behavior. Those who answered "Yes, but it's very hard for me," and "Yes, and it's very easy for me," were considered as such. However, in Table 2 the numbers of study participants with the detailed statements are reported, i.e., the number of individuals stating "No, I do not intend to," (1), "No, but I've thought about it," (2), "No, but I've decided to do it," (3), "Yes, but

it's very difficult for me," (4), and "Yes, and it's very easy for me," (5) with hand hygiene and face mask wearing, differentiated for the overall sample, the general population and the patients.

**Table 2.** Prevalence of hand hygiene and face mask wearing (number of study participants, and percentages in parentheses; missing numbers to 100% are due to non-responses to these questions).

| Hand Hygiene | Overall | General Population | Patients [0] |
|---|---|---|---|
| No, I do not intend to. | 442 (11.0%) | 404 (21.5%) | 38 (1.8%) |
| No, but I've thought about it. | 181 (4.5%) | 128 (6.8%) | 53 (2.5%) |
| No, but I've decided to do it. | 116 (2.9%) | 63 (3.4%) | 53 (2.5%) |
| Yes, but it's hard for me. | 680 (17.0%) | 266 (14.2%) | 414 (19.4%) |
| Yes, and it's easy for me. | 2589 (64.6%) | 1014 (54.1%) | 1575 (73.8%) |
| **Face Mask Wearing** | **Overall** | **General Population** | **Patients [0]** |
| No, I do not intend to. | 111 (2.8%) | 108 (5.7%) | 3 (0.1%) |
| No, but I've thought about it. | 8 (0.2%) | 7 (0.4%) | 1 (0.1%) |
| No, but I've decided to do it. | 2 (0.1%) | 2 (0.1%) | 0 (0.0%) |
| Yes, but it's hard for me. | 1426 (35.4%) | 828 (44.0%) | 598 (27.8%) |
| Yes, and it's easy for me. | 2483 (61.5%) | 936 (49.8%) | 1547 (72.0%) |

[0] patients with mental health problems admitted to a medical rehabilitation.

For hand hygiene, the differences between the general population and the patients were significant with $Chi^2$ (df = 4) = 474.137; $p < 0.001$. For face mask wearing, the differences between the general population and the patients were significant with $Chi^2$ (df = 4) = 276.673; $p < 0.001$.

According to overall self-reports of avoiding large masses and keeping the physical distance of 1.5 m to others, 88.0% and 88.1% of the study participants were performing the recommended hygiene behavior. Those who answered: "Agree to some extent" and "Agree fully" were considered as compliant. Individuals answering "No, I do not intend to,", "No, but I've thought about it," and "No, but I've decided to do it," were classified as non-compliant. Accordingly, in Table 3 the numbers of study participants with the detailed statements are reported, i.e., the number of individuals stating that they would "Do not agree at all" (1), "Do rather not agree" (2), "Agree to some extent" (3), or "Agree fully" with avoiding large masses and keeping the distance to others (Table 3).

**Table 3.** Prevalence of avoiding large masses and keeping distance (number of study participants, and percentages in parentheses; missing numbers to 100% are due to non-responses to these questions).

| Avoiding Large Masses | Overall | General Population | Patients [0] |
|---|---|---|---|
| Do not agree at all | 232 (6.0%) | 220 (12.5%) | 12 (0.6%) |
| Do rather not agree | 238 (6.1%) | 198 (11.2%) | 40 (1.9%) |
| Agree to some extent | 1210 (31.1%) | 562 (31.9%) | 648 (30.4%) |
| Agree fully | 2216 (56.9%) | 784 (44.4%) | 1432 (67.2%) |
| **Keep the Physical Distance of 1.5 m to Others** | **Overall** | **General Population** | **Patients [0]** |
| Do not agree at all | 143 (3.7%) | 132 (7.5%) | 11 (0.5%) |
| Do rather not agree | 321 (8.2%) | 284 (16.1%) | 37 (1.7%) |
| Agree to some extent | 1901 (48.7%) | 872 (49.4%) | 1029 (48.2%) |
| Agree fully | 1536 (39.4%) | 476 (27.0%) | 1060 (49.6%) |

[0] patients with mental health problems admitted to a medical rehabilitation.

For avoiding large masses, the differences between the general population and the patients were significant with $Chi^2$ (df = 3) = 456.284; $p < 0.001$. For keeping the physical distance of 1.5 m to others, the differences between the general population and the patients were significant with $Chi^2$ (df = 3) = 496.325; $p < 0.001$.

Aggregating all four behaviors to one measure indicating how many behaviors are performed as recommended, differences were also investigated between patients and the general population (Figure 1).

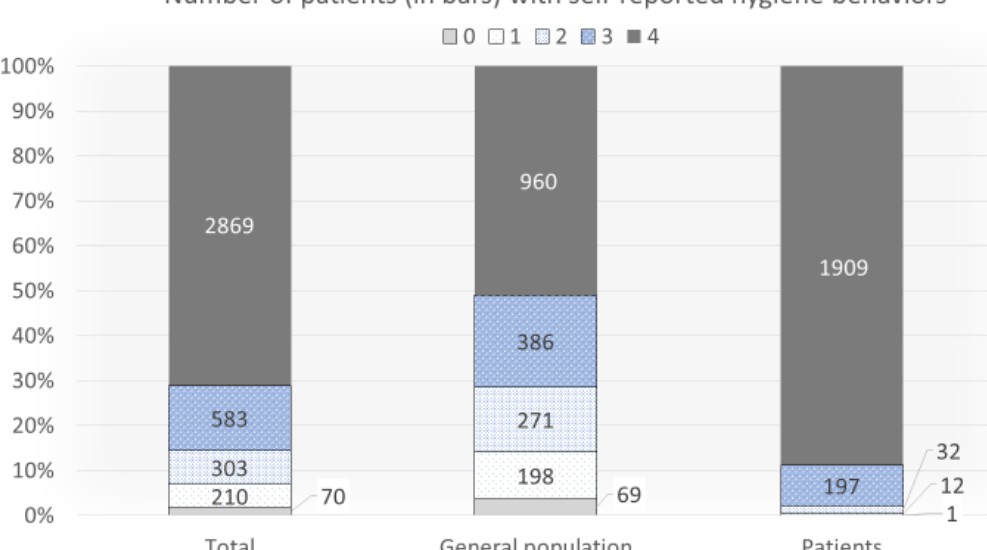

**Figure 1.** Prevalence's for all four hygiene behaviors aggregated as an indicator for self-reported compliance within the total sample and differentiated for the general population and patients with mental health problems admitted to a medical rehabilitation (Chi$^2$ (df = 4) = 780.246; $p < 0.01$).

It revealed that the patient group was much more compliant according to self-reported behavior than the general population, with more than 80% performing all four behaviors compliantly, whereas this rate was much smaller with only about 50% of the general population, and about 70% in both samples combined (Figure 1).

*3.2. Research Question 2: To What Extent Are Hygiene Behaviors (Avoiding Masses, Physical Distancing, Hand Hygiene, and Face Mask Use) Correlated with Each Other Overall and in the Two Samples?*

The interrelations of the different behaviors are reported in Table 4.

**Table 4.** Interrelations of the different hygiene behaviors with each other (Spearman's rho; above the diagonal the patient sample, below the diagonal represents the general population, with both groups aggregated in parentheses).

| | Hand Hygiene | Face Mask Wearing | Avoiding Large Masses | Keep the Distance | Aggregate of All Behaviors |
|---|---|---|---|---|---|
| Hand hygiene | 1 | 0.240 ** | 0.150 ** | 0.169 ** | 0.457 ** |
| Face mask wearing | 0.405 ** (0.368 **) | 1 | 0.091 ** | 0.084 ** | 0.084 ** |
| Avoiding large masses | 0.447 ** (0.364 **) | 0.502 ** (0.360 **) | 1 | 0.498 ** | 0.266 ** |
| Keep the physical distance of 1.5 m to others | 0.420 ** (0.353 **) | 0.432 ** (0.314 **) | 0.693 ** (0.628 **) | 1 | 0.263 ** |
| Aggregate of all behaviors | 0.647 ** (0.710 **) | 0.490 ** (0.600 **) | 0.717 ** (0.752 **) | 0.671 ** (0.759 **) | 1 |

** $p < 0.001$.

The interrelations with both groups aggregated range from 0.31 to 0.63, with the highest interrelations between avoiding large masses and keeping a physical distance of

1.5 m from others, whereas all other correlations are around 0.36. All correlations are statistically significant, indicating that individuals who perform one hygiene behavior are more likely to also perform the other hygiene behaviors (and rather not compensate for each other).

The correlations are much higher in the general population than in the patient group (Table 4).

### 3.3. To What Extent Are Hygiene Behaviors (Avoiding Masses, Physical Distancing, Hand Hygiene, and Face Mask Use as Well as the Aggregated Behaviors) Correlated with Overall Fear in the Two Samples?

The interrelations with both groups aggregated are reported in Table 5 showing that correlations range from 0.52 to 0.44, with correlations being statistically significant. The correlations are much higher in the general population than in the patient group (Table 5).

**Table 5.** Interrelations between hygiene behaviors and fears Spearman's rho (first correlate coefficient for patients, second for general population and both groups aggregated in parentheses).

| | Hand Hygiene | Face Mask Wearing | Avoiding Large Masses | Keep the Distance of 1.5 m to Others | Aggregate of All Behaviors |
|---|---|---|---|---|---|
| Fear of being infected with the coronavirus | 0.033/0.368 ** (0.258 **) | −0.086 **/0.441 ** (0.235 **) | 0.204 **/0.566 ** (0.432 **) | 0.240 **/0.469 ** (0.394 **) | 0.175 **/0.496 ** (0.413 **) |
| Fear of getting seriously ill with COVID-19 | 0.051 */0.321 ** (0.248 **) | −0.103 **/0.393 ** (0.207 **) | 0.186 **/0.525 ** (0.410 **) | 0.226 **/0.446 ** (0.386 **) | 0.170 **/0.446 ** (0.399 **) |
| Fear of infecting relatives/roommates or friends with COVID-19 | 0.055 */0.330 ** (0.230 **) | −0.051 */0.407 ** (0.218 **) | 0.141 **/0.463 ** (0.333 **) | 0.164 **/0.386 ** (0.298 **) | 0.147 **/0.441 ** (0.345 **) |
| Aggregate of all fears | 0.056 */0.379 ** (0.273 **) | −0.094 **/0.470 ** (0.246 **) | 0.196 **/0.578 ** (0.435 **) | 0.233 **/0.483 ** (0.398 **) | 0.189 **/0.517 ** (0.430 **) |

** $p < 0.001$; * $p < 0.05$.

Interrelations were also tested beyond the linear trend. Therefore, the four hygiene behaviors were aggregated and the three fear items were averaged to test for general patterns. With the aggregated hygiene behavior and the aggregated fear index, it revealed that not only linear interrelations ($\beta_{linear} = 3.355$; $p < 0.001$) explain the variance, but also the quadratic term ($\beta_{quadratic} = -4.953$; $p < 0.001$) and the cubic terms ($\beta_{cubic} = 2.102$; $p < 0.001$) were significant, and explained 29.8% of the variance. The trends are displayed in Figure 2.

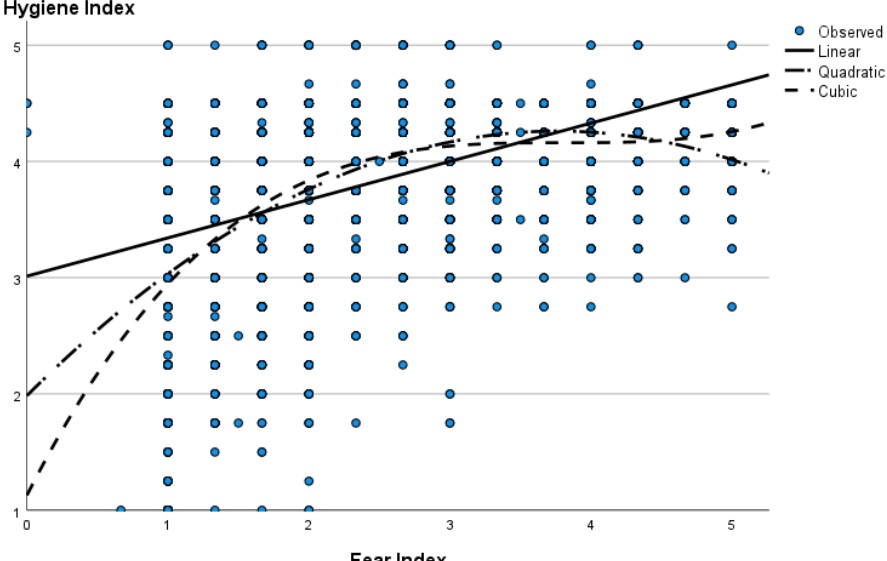

**Figure 2.** Interrelation of fear relating to a COVID-19 infection and hygiene behaviors.

The same pattern was revealed with all single behaviors and two fear items, namely fear of being infected with SARS-CoV-2 and fear of getting seriously ill with COVID-19. However, concerning the item fear of infecting relatives/roommates or friends with COVID-19 it revealed a different pattern: although fear was significantly interrelated with hand hygiene, avoiding masses, and keeping a distance in a linear and quadratic way ($\beta_{linear}$ = 1.043–1.313; $p < 0.001$; $\beta_{quadratic}$ = −1.087 to −1.473; $p$ = 0.003–0.030) the cubic trend was not significant ($\beta_{cubic}$ = 0.277–0.527; $p$ = 0.065–0.343). Only wearing a face mask was significantly interrelated with fear in a linear, quadratic, and cubic fashion ($\beta_{linear}$ = 1.568; $p < 0.001$; $\beta_{quadratic}$ = −2.006; $p < 0.001$; $\beta_{cubic}$ = 0.699; $p$ = 0.017).

Although this effect was stronger in the general population than in the patient sample. In the patient group, the linear and the quadratic effects were significant ($\beta_{linear}$ = 1.169; $p < 0.001$; $\beta_{quadratic}$ = −1.729; $p$ = 0.028) but the cubic terms were not significant ($\beta_{cubic}$ = 1.755; $p$ = 0.079) and only 4.6% of the variance could be explained. In the general population the linear, the quadratic effects and the cubic terms were significant ($\beta_{linear}$ = 2.487; $p < 0.001$; $\beta_{quadratic}$ = −2.984; $p < 0.001$; $\beta_{cubic}$ = 1.054; $p < 0.001$) and 36.5% of the variance could be explained.

Lastly, the means of the different compliances regarding self-reported hygiene behaviors were analyzed in a MANOVA controlling for group. Although fear was significantly different for the different self-reported compliance rates ($F_{Wilks' Lambda}$ (3, 9526) = 32.813; $p < 0.001$; $Eta^2$ = 0.024) and for the two groups ($F_{Wilks' Lambda}$ (3, 3914) = 3.820; $p$ = 0.01; $Eta^2$ = 0.003), there was no interaction ($F_{Wilks' Lambda}$ (9, 9526) = 1.254; $p$ = 26; $Eta^2$ = 0.001). The means are reported in Figure 3.

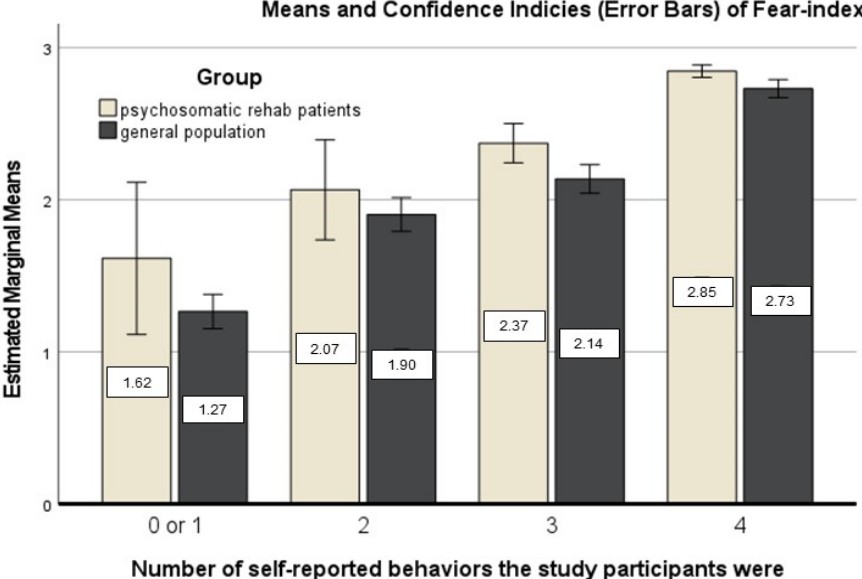

**Figure 3.** Mean (and CIs) scores of fear (aggregated from fear of being infected with the coronavirus, fear of getting seriously ill with COVID-19 and fear of infecting relatives/roommates or friends with COVID-19) within compliance groups regarding self-reported hygiene behaviors.

The trend clearly demonstrates that the more hygiene behaviors are complied with, the higher the fear levels, and psychosomatic patients have a higher proportion of compliance than the general population.

## 4. Discussion

The current study aimed to examine the associations between hygiene behaviors (i.e., keeping distance, avoiding masses, wearing face masks and complying with hand hygiene recommendations) during the COVID-19 pandemic with COVID-related fear in the general population as well as a sample of persons with mental health symptoms by applying the CCAM.

Regarding *research question 1*, the highest self-reported compliance rates in the overall population (individuals performing the recommended hygiene behavior) could be detected with face mask wearing followed by hand hygiene. Avoiding large masses and keeping a physical distance of 1.5 m was performed consequently by fewer individuals. Summarizing, approximately 82% to 97% of participants can be considered mostly compliant. With face mask wearing, the number of non-compliant participants was small, at just 3%. However, there was a surprisingly large number of non-compliant individuals in the overall population regarding hand hygiene (18%) who reported that they did not complying with hand hygiene recommendations and also had no intention to consider it (indicated "No, I do not intend to"). This finding reflects the findings of another report on infection prevention via hand hygiene. The authors found that approximately 18% of all participants washed their hands less than five times a day. Important barriers included a perceived lack of necessity or adequate hand hygiene options, forgetting to wash hands or sometimes time constraints [36].

Summarizing the group differences concerning the first research question, we found main differences between patients and the general population in the compliance to hygiene behaviors. Psychosomatic rehabilitation patients were much more compliant with 80% in comparison to only 50% compliant individuals in the general population. A possible explanation for the results may be the primary diagnosis according to the International Classification of Disease-10 Manual (ICD-10). Most patients from the rehabilitation clinics were diagnosed with either an affective disorder or an anxiety disorder. Due to the nature of those, patients usually perceive more worries and fears associated with uncertain situations which could also affect their corresponding behavior [15,20,24,26]. Referring to another report [36], this is a new barrier that could have developed during the course of the COVID-19 pandemic and should be considered in the future, especially when communicating the need for hygiene behavior performance. Accordingly, communication is required to ensure high hygiene standards and patient safety, and to prevent adverse effects such as too-elevated fear and risk perception.

There is a lack of compliance to COVID-19 containment measures in the general population, especially regarding hand hygiene. This has been confirmed before in healthcare workers and the general public [37,38]. Reasons can be a lack of knowledge and awareness, but also other behavioral determinants [39]. However, the lack of hand hygiene can increase the spread of the SARS-CoV-2 virus; therefore, there is a high need for adequate hand hygiene measures in the general population. The CCAM model suggests that planning and self-efficacy can help to overcome the intention–behavior gap, which in turn can prove beneficial not only for one hygiene behavior (such as hand hygiene), but also other hygiene behaviors due to carry-over mechanisms [40]. Although each behavior must be intended and controlled individually, they also change as a result of one another. This can be promising on the one hand since a positive change in one behavior can be transferred to other behaviors. On the other hand, these carry-over mechanisms can pose a threat since non-compliance to one behavior (e.g., wearing a face mask) can negatively impact other behaviors (e.g., avoiding mass gatherings), thus leaving the individual vulnerable for negative outcomes, i.e., the increased risk of COVID-19 infections in this case.

What is promising in terms of finding a comprehensive strategy is our result that COVID-19 hygiene behaviors were intercorrelated in our study (*research question 2*), especially with regard to the two physical distancing measures of avoiding masses and keeping a distance of 1.5 m between humans, which showed a large correlation. This supports the idea that the CCAM model could inform compliance interventions. The correlation between hand hygiene, wearing a face mask and the physical distancing measures appeared slightly lower than avoiding large masses and keeping the distance. However, the correlations were higher in the general population than they were in the psychosomatic rehabilitation patients. This may indicate more transfer in the general population but also the lower variance in the patient sample. Further research is needed to investigate compensation and what holds individuals from compliance.

Answering *research question 3*, in the psychosomatic rehabilitation group, COVID-19 related fears were less strongly associated with hygiene behaviors than in the general population. Different mental health diagnoses of the psychosomatic rehabilitation patients may provide an explanation for the results. Being susceptible to anxiety may consequently lead to more fears and worries associated with infecting others [34]. On the contrary, participants from the general population may display a wide range of fear and hygiene behavior (i.e., more variance to be explained), leading to higher associations.

All hygiene behaviors were related to the fear of spreading, infecting oneself or getting seriously ill with COVID-19 in a linear and quadratic way. This indicates that after a peak, the likelihood of performing hygiene behaviors does not increase anymore. Rather, there seems to be a medium level working best, whereas more fear is rather dysfunctional. This finding has been termed the Yerkes-Dodson Law [28], and can be found in all areas of behavior change. Regarding the COVID-19 pandemic, individuals with high levels of fear are less likely to actively engage in behavior change. This can be described well in terms of mask wearing: disengaging in mask wearing and attending mass gatherings can signal a certain "normality" that people are looking for to reduce their stress levels.

On the other hand, the pandemic has been going on for 2 years already now. Thus, seeing masks and reminders to wash or disinfect hands have become more normal and individuals have become more used to pandemic cues which they might even perceive as "cue-to-action" [41]. Hand hygiene and other hygiene behaviors have been required in prior flu seasons; nevertheless, containment measures for COVID-19 are a lot more comprehensive and thus complicated. Additionally, containment measures have changed with regard to new knowledge (e.g., wearing masks outside was not required in Germany until May 2020 but was then introduced in city centers and other frequently crowded areas depending on infection rates). There is a high need to practice all behaviors adequately and to find a good, individual strategy incorporating all hygiene behaviors into daily life [41]. In the COVID-19 Snapshot Monitoring (COSMO), the authors state that the current protection behavior has increased during the fourth wave of COVID-19 in Germany. This indicates that the general population adapts their behavior to current risks even though the compliance rates were higher during the third wave. The increase in protection behavior was associated with increased worries which reflects our current findings that COVID-related fears are associated with hygiene behavior [42] but not only linearly but also quadratically.

To address fears, create adequate risk perceptions, and thus foster hygiene behaviors, more effort in research and practice needs to be focused on public risk communication. It has been established that the ongoing presence in social media, including all stakeholders and addressing risk perceptions can be helpful to target disruptive behavior over the course of a pandemic [43,44]. The World Health Organization has strongly recommended to incorporate *risk communication and community engagement* (RCCE) in public health emergency plans in order to prevent "infodemics" and build trust so that social disruption can be avoided and effective response can be created [45]. Addressing competence perception and self-efficacy believes may as important or even more effective as outcome expectancies which may result in functional or too elevated fear.

Although our study sheds light on the association of hygiene behavior, COVID-19 related fear and higher-level goals, several *limitations* must be borne in mind. Firstly, we used self-reported data to assess compliance with hygiene measures. However, especially research regarding healthcare workers' compliance to hand hygiene has found that self-reports can be unreliable compared with standardized observations (in combination with product use monitoring) [46,47]. Due to social desirability, cognitive dissonance and self-serving bias, self-reports tend to overestimate actual compliance to hygiene behaviors [48]. Hence, it is likely that we overestimated the actual compliance to hygiene behaviors. Furthermore, non-random sampling procedures in both surveys is a major limitation of the study, including the fact that no survey response rates could be reported. There might be associated probable critical selection biases, which needs to be considered.

Our results should be interpreted with caution when looking at percentages and compliance rates. Future research should replicate our study in an observational setting. Since the general population was assessed online and the sample of rehabilitation patients was recruited at four rehabilitation clinics, they might not be representative of the German population. Additionally, the samples were recruited between May 2020 and August 2021 which is a long time during the dynamic pandemic situation. We did not control for fear and hygiene differences between COVID-19 waves and periods with lower incidences. Our study examined the associations of COVID-19 related fears and hygiene behaviors in a cross-sectional dataset. Hence, causality changes over time could not be evaluated. In future research, more longitudinal studies as well as intervention studies [1] and more comparisons of different theories as well as more synchronized measures are required (i.e., all hygiene behaviors should be assessed with the same answering format) [1,31]. Finally, the study was conducted with German samples only so that results might differ between countries and cultural contexts because COVID-19 containment measures were highly heterogeneous when regarding different contexts and regions.

Nevertheless, our study has several *practical implications*. Generally, individuals were more self-reported compliant if adequately aware of risks. Consequently, the public and patients should be educated well without inducing excessive levels of fear. Clear, objective *risk communication and community engagement* should be applied by educating about risks while offering clear behavioral recommendations and strengthening self-efficacy. How to promote higher-level goals even in face of restrictions and to stay physically and mentally healthy could be emphasized. Risk communication may not only be carried out by health professionals, but also by media and authorities, thus containing the spread of the SARS-CoV-2 virus. When communicating risks, controllability, in terms of the need for performing hygiene behavior and how to execute them, needs to be incorporated and too-elevated fear should be prevented.

## 5. Conclusions

In summary, we conclude that individuals who feel more vulnerable to fall (severely) ill from COVID-19 and psychosomatic rehabilitation patients are more compliant, according to their self-reports. However, there is a (relatively) high percentage of self-reported non-compliant individuals in the general population, particularly in regard to hand hygiene measures. To promote hygiene behaviors, individuals should be carefully educated about risks ensuing from their behavior, so that a medium level of fear can be reached. If levels of fear of infecting oneself and others or becoming seriously ill are either too low or too high, individuals will tend to wither disregard or avoid hygiene behaviors. The need and execution for hygiene behaviors should be communicated carefully also to functionally buffer the risk of being exposed to infections.

**Author Contributions:** Conceptualization, A.D., S.L., and F.M.K.; methodology, S.L.; software, F.M.K., S.L., and A.D.; validation, A.D., S.L., and C.D.; formal analysis, S.L.; investigation, S.L., A.D., and F.M.K.; resources, A.D., and S.L.; data curation, F.M.K.; writing—original draft preparation, S.L.; writing—review and editing, F.M.K., A.D., C.D., and L.K.; visualization, S.L.; supervision, A.D.; project administration, S.L., F.M.K., and A.D.; funding acquisition, S.L., and A.D. All authors have read and agreed to the published version of the manuscript.

**Funding:** This research was partially funded by the German Innovation Fund (Project No. 01VSF18023) of The Federal Joint Committee (G-BA) as part of the research project "TeamBaby-Safe, digitally supported communication in obstetrics and gynecology" (grant 01VSF18023). The funding body only observes whether the study is outperformed as applied and approved in terms of the design of the study. The funding body has no influence on the collection, analysis, and interpretation of data. The funding body is not involved in writing the manuscripts and will be informed about the authors' activities.

**Institutional Review Board Statement:** The study was conducted according to the guidelines of the Declaration of Helsinki and approved by the Ethics Committee of Jacobs University Bremen (protocol code 2020_09 and date of approval: 25 June 2020).

**Informed Consent Statement:** Informed consent was obtained from all subjects involved in the study.

**Data Availability Statement:** The data presented in this study are available on request from the corresponding author. The data are not publicly available due to confidential patient data being used.

**Acknowledgments:** We would like to thank the Becker clinics Möhnesee, Norddeich, Juliana, and Burg for their assistance in data collection. We also appreciate the support of Kureva Matuku by performing parts of the literature search to review the current state of the science. Furthermore, we would like to thank Ronja Bellinghausen for proofreading this manuscript.

**Conflicts of Interest:** The authors declare no conflict of interest.

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
