# Peer review of "Hygiene Behaviors and SARS-CoV-2-Preventive Behaviors in the Face of the COVID-19 Pandemic: Self-Reported Compliance and Associations with Fear, SARS-CoV-2 Risk, and Mental Health in a General Population vs. a Psychosomatic Patients Sample in Germany"

_2673-947X, doi:10.3390/hygiene2010003_

Round 1

Reviewer 1 Report

The authors report survey data on COVID-19-preventive behaviors from two studies, an online general population survey and an online survey of patients before medical rehabilitation. They report tests related to an array of research questions, not all of which are, to me, convincing and sufficiently connected to one another. Thus, as an original research article, this paper wants too much, so I definitely suggest the following "re-configuration" of the data:

1.) Research Questions 1(a) and 2(i) should become Research Question 1: prevalences of behaviors, overall and in the two samples in comparison; 

2.) Research Question 3 should become Research Question 2: correlations between behaviors, and prevalences of combined/aggregated behavior/s, overall and in the two samples.

3.) Research Questions 1(b) and 2(ii) should become Research Question 3: relationships of the single and combined/aggregated behavior(s) to fears, overall and in the two samples in comparison.

4.) Research Questions 1(c) and 4 should be omitted: To me, the former is trivial (differences between population and patients in health?!?), and the latter not convincingly delineated, and yields results that the authors tend to over-interpret (e.g., correlations of 0.048 as indicative of any notable relation between variables). Likewise, Research Question 5 should also be omitted as its delineation and interpretation seem to me rather far-fetched, and...

... the paper still has more than enough to offer when presenting 1.)-3.), i.e.:

1.) estimates of prevalances of self-reported individual COVID-19 preventive behaviors during the pandemic in a general population and a patient sample,

2.) correlations between these self-reported behaviors and estimates of prevalances of multiple behaviors indicators during the pandemic in a general population and a patient sample, and

3.) relations of the above with fear.

When doing so, and revising their paper, the authors should consider the following:

1.) As an indicator of multiple behaviors, not only a mean score should be analysed, but also a variable of how many behaviors were self-reportedly compliant (see the methodology in ref. [4] on hand washing attributes for an analogous example). This would allow to report prevalences for more than one (and up to four) COVID-19 behaviors.

2.) The authors should definitely discuss their results in comparison to other studies such as the COSMO (https://projekte.uni-erfurt.de/cosmo2020/web) and earlier data based on the BZgA surveys on hygiene and infection control (https://www.bzga.de/forschung/studien/abgeschlossene-studien/studien-ab-1997/impfen-und-hygiene, e.g. [4]), i.e. in order to help readers compare.

3.) Finally, it really made me wonder why the authors did not once wonder whether self-reported behaviors may assess NOT compliance, but OVER-estimation of compliance - and with this a presumed determinant of NON-compliance (that is, the more I overestimate, the more I falsely think everything's fine, and no actions are needed). Thus, I definitely recommend in-depth discussion of whether and to what extent (if at all) self-reports of behavior represent actual observable behavior. This also implies that the paper should state to report prevalences of self-reported compliance, not compliance.

MINOR POINTS:

1.) I am not aware that reference [4] contains significant content justifying its mentioning on p 7, line 46. However, it is relevant (see above).

2.) Figures 1-2: Please check the descriptions on the x-axis: shouldn't it be "neither disabled nor risk factors", "disabled or at least one risk factor", and "both disabled and at least one risk factor"? Also, specify the risk factors as a note.

3.) Table 3: I guess behaviors are in here as scales (1-5). Epidemiologically, it would be at least as interesting to either see the means of the fear scales by compliance vs. non-compliance or the probability of compliance given fear vs. less/no fear. In other words, infection prevention and control audiences are less acquainted to correlations, and means + % tend to be more telling.

Author Response

Review 1

The authors report survey data on COVID-19-preventive behaviors from two studies, an online general population survey and an online survey of patients before medical rehabilitation. They report tests related to an array of research questions, not all of which are, to me, convincing and sufficiently connected to one another. Thus, as an original research article, this paper wants too much, so I definitely suggest the following "re-configuration" of the data:

1.) Research Questions 1(a) and 2(i) should become Research Question 1: prevalences of behaviors, overall and in the two samples in comparison; 

2.) Research Question 3 should become Research Question 2: correlations between behaviors, and prevalences of combined/aggregated behavior/s, overall and in the two samples.

3.) Research Questions 1(b) and 2(ii) should become Research Question 3: relationships of the single and combined/aggregated behavior(s) to fears, overall and in the two samples in comparison.

4.) Research Questions 1(c) and 4 should be omitted: To me, the former is trivial (differences between population and patients in health?!?), and the latter not convincingly delineated, and yields results that the authors tend to over-interpret (e.g., correlations of 0.048 as indicative of any notable relation between variables). Likewise, Research Question 5 should also be omitted as its delineation and interpretation seem to me rather far-fetched, and...

... the paper still has more than enough to offer when presenting 1.)-3.), i.e.:

1.) estimates of prevalances of self-reported individual COVID-19 preventive behaviors during the pandemic in a general population and a patient sample,

2.) correlations between these self-reported behaviors and estimates of prevalances of multiple behaviors indicators during the pandemic in a general population and a patient sample, and

3.) relations of the above with fear.

RESPONSE BY THE AUTHORS: Thank you very much for this constructive feedback. We accordingly changed the research questions and trimmed the text. Accordingly, we omitted the focus on disability and life-satisfaction. The changes research question now read as follows (page 3 to 4):

Research Question 1: What is the prevalence of hygiene behaviors, overall and in the two samples in comparison general population vs. patients with mental health problems?

Research Question 2. To what extent are hygiene behaviors (avoiding masses, physical distancing, hand hygiene and face mask use as well as the aggregated behaviors) correlated with each other overall and in the two samples?

Research Question 3. To what extent are hygiene behaviors (avoiding masses, physical distancing, hand hygiene and face mask use as well as the aggregated behaviors) correlated with overall fear in the two samples?

When doing so, and revising their paper, the authors should consider the following:

1.) As an indicator of multiple behaviors, not only a mean score should be analysed, but also a variable of how many behaviors were self-reportedly compliant (see the methodology in ref. [4] on hand washing attributes for an analogous example). This would allow to report prevalences for more than one (and up to four) COVID-19 behaviors.

RESPONSE BY THE AUTHORS: Thanks for this constructive suggestion. We accordingly added such an aggregated score: Please see Table 2 and 3 as well as Figure 1.

2.) The authors should definitely discuss their results in comparison to other studies such as the COSMO (https://projekte.uni-erfurt.de/cosmo2020/web) and earlier data based on the BZgA surveys on hygiene and infection control (https://www.bzga.de/forschung/studien/abgeschlossene-studien/studien-ab-1997/impfen-und-hygiene, e.g. [4]), i.e. in order to help readers compare.

RESPONSE BY THE AUTHORS: Thank you very much for this feedback. We accordingly added references and trimmed the text as follows:

“This finding reflects the findings of the BZgA report on infection prevention via hand hygiene. The authors found that approximately 18% of all participants washed their hands less than five times a day. Important barriers included a perceived lack of necessity or adequate hand hygiene options, forgetting to wash hands or sometimes time constraints [36].

Summarizing the group differences concerning the first research question, we found main differences between patients and the general population in the adherence to hygiene behaviors. Psychosomatic rehabilitation patients were about twice as compliant. A possible explanation for the results may be the primary diagnosis according to the International Classification of Disease – 10 Manual (ICD-10). Most patients from the rehabilitation clinics were diagnosed with either an affective disorder or an anxiety disorder. Due to the nature of those, patients usually perceive more worries and fears associated with uncertain situations which could also affect their corresponding behavior [46,47]. Referring to the BZgA report, this is a new barrier which could have developed during the course of the COVID-19 pandemic and should be considered in the future especially when communicating the need of hygiene behavior performance. According communication is required to ensure high hygiene standards and patient safety, and to prevent adverse effects such as too elevated fear and risk perception.” (line 407ff).

“There is a high need to practice all behaviors adequately and to find a good, individual strategy incorporating all hygiene behaviors into daily life [41]. In the COVID-19 Snapshot Monitoring (COSMO), the authors state that the current protection behavior has increased during the fourth wave of COVID-19 in Germany. This indicates that the general population adapts their behavior to current risks even though the adherence rates were higher during the third wave. The increase in protection behavior was associated with increased worries which reflects our current findings that COVID-related fears are associated with hygiene behavior [42] but not only linearly but also quadratically.” (lines 472ff).

3.) Finally, it really made me wonder why the authors did not once wonder whether self-reported behaviors may assess NOT compliance, but OVER-estimation of compliance - and with this a presumed determinant of NON-compliance (that is, the more I overestimate, the more I falsely think everything's fine, and no actions are needed). Thus, I definitely recommend in-depth discussion of whether and to what extent (if at all) self-reports of behavior represent actual observable behavior. This also implies that the paper should state to report prevalences of self-reported compliance, not compliance.

RESPONSE BY THE AUTHORS: We appreciate this suggestion and accordingly we added self-reported to compliance and trimmed the discussion regarding in-depth discussion of whether and to what extent (if at all) self-reports of behavior represent actual observable behavior:

“While our study sheds light on the association of hygiene behavior, COVID-19 related fear and higher-level goals, several limitations must be borne in mind. Firstly, we used self-reported data to assess adherence to hygiene measures. However, especially research regarding healthcare workers’ adherence to hand hygiene has found that self-reports can be unreliable compared to standardized observations (in combination with product use monitoring) [46,47]. Due to social desirability, cognitive dissonance and self-serving bias, self-reports tend to over-estimate actual adherence to hygiene behaviors [48]. Hence, it is likely that we over-estimated the actual adherence to hygiene behaviors. Our results should be interpreted with caution when looking at percentages and adherence rates. Future research should replicate our study in an observational setting.” (lines 489ff).

MINOR POINTS:

1.) I am not aware that reference [4] contains significant content justifying its mentioning on p 7, line 46. However, it is relevant (see above).

RESPONSE BY THE AUTHORS: Thanks for this feedback and apologize for the incorrect use of this reference. We have consequently exchanged the reference with a new reference, the German Law of Infection prevention Law, Section 28a and 28b [4]. (lines 48f).  

2.) Figures 1-2: Please check the descriptions on the x-axis: shouldn't it be "neither disabled nor risk factors", "disabled or at least one risk factor", and "both disabled and at least one risk factor"? Also, specify the risk factors as a note.

RESPONSE BY THE AUTHORS: We appreciate this point. However, as we removed the former Figures 1, 2 and 3 due to the feedback regarding the research questions, this is no concern any more.

3.) Table 3: I guess behaviors are in here as scales (1-5). Epidemiologically, it would be at least as interesting to either see the means of the fear scales by compliance vs. non-compliance or the probability of compliance given fear vs. less/no fear. In other words, infection prevention and control audiences are less acquainted to correlations, and means + % tend to be more telling.

RESPONSE BY THE AUTHORS: Thank you, we agree and accordingly, we added matching analyses:

“Lastly, the means of the different compliances regarding self-reported hygiene behaviors were analyzed in a MANOVA controlling for group. While fear was significantly different for the different self-reported compliances rates (FWilks' Lambda(3, 9526)=32.813; p<.001; Eta²=.024) and for the two groups (FWilks' Lambda(3, 3914)=3.820; p=.01; Eta²=.003), there was no interaction (FWilks' Lambda(9, 9526)=1.254; p=26; Eta²=.001). The means are reported in Figure 3.

Figure 3. Mean (and CIs) scores of fear (aggregated from fear of being infected with the coronavirus, fear of getting seriously ill with COVID-19 and fear of infecting relatives/roommates or friends with COVID-19) within compliance groups regarding self-reported hygiene behaviors.

The trend clearly demonstrates, that the more hygiene behaviors are complied with, the higher the fear levels.“ (lines 378ff).

Reviewer 2 Report

Overall, your manuscript looks important, but it's impossible to read, especially for the average reader. There are 27 pages, a series of unnecessary information and 6 tables and 4 figures, when the normal is 5 adding the two. In the current format, the chances of reading and quoting the manuscript are small.
Please objectively focus on the interrelationships between hygiene-related behaviors and return your manuscript for further review.

Author Response

Review 2

Overall, your manuscript looks important, but it's impossible to read, especially for the average reader. There are 27 pages, a series of unnecessary information and 6 tables and 4 figures, when the normal is 5 adding the two. In the current format, the chances of reading and quoting the manuscript are small.
Please objectively focus on the interrelationships between hygiene-related behaviors and return your manuscript for further review.

RESPONSE BY THE AUTHORS: Thanks for this feedback and we are sorry for the challenges with reading the paper. Following the suggestion of Reviewer 1, we omitted two research questions and excluded related analyses and results. At the same time, we reduced the number of tables to 5 and of figures to 3. In case you still think, this is too much, we could also move Table 1 and Figure 3 as well as parts of the manuscript to the appendix (the former appendix was deleted).

The manuscript is now only 19 pages long (of which 4 are only references).

Reviewer 3 Report

An interesting paper, but a more thorough background on the literature of risk communication would be helpful. Also, what exactly do the authors mean by "fear". This does not appear to be explained and would benefit from being unpacked and operationalized for the reader to understand the authors' intent. In general, the discussion appears to have more speculation, than interpretation through a theoretical lens, and it would be recommended that this, in particular, be revised.

A thorough copy edit would be needed to strengthen the writing and grammar.

Author Response

Review 3

An interesting paper, but a more thorough background on the literature of risk communication would be helpful. Also, what exactly do the authors mean by "fear". This does not appear to be explained and would benefit from being unpacked and operationalized for the reader to understand the authors' intent.

RESPONSE BY THE AUTHORS: Thank you for this request. We improved the introduction accordingly. However, we would like to point out that as part of the extensive revision process, we have removed risk perception from the analyses. Therefore, we have discussed the term “fear” more extensively in the following:

„…it was often found that the level of fear (negative emotion relating to anxiety, worry or concern of own risk) of an infection with the SARS-CoV-2 virus was interrelated with the prevalence of frequent hand hygiene [7,8] but not with physical distancing, leaving home for food or to exercise [8]. Also, the fear of becoming seriously ill with COVID-19 was correlated with behavior change [9]. Vulnerability, perceived risk, and fear were all found to be significantly interrelated with preventive behaviors during the COVID-19 pandemic [10], but results are not always consistent [3]. It has been found that fear can be buffered by hand hygiene behavior [11] and thereby work functionally [9]. However, whether this is the case with fear and other hygiene behaviors, too, has not been systematically tested so far relating to the COVID-19 pandemic in Germany.“ (lines 52ff). and

“COVID-19 related fear and behavior

Fear, as an adaptive response to a perceived threat or danger, has been reported to be the first emotional response to the COVID-19 pandemic. Thus, fear of an infection and possible consequences can either be irrational or rational and commonly concern getting infected, transmitting the disease, losing relatives or possible negative health outcomes (i.e. post-COVID) [18,19]. While people with risk factors for COVID-19 have received considerable early attention [2,20], less attention has been paid to how other susceptible populations i.e. individuals with previous mental conditions cope with the fear of a COVID-19 infection [21]. Although increasingly systematic research is now available on risk factors causing a severe course of COVID-19 (like cardiovascular disease, diabetes, respiratory illnesses, liver disease, kidney disease or cancer) [22], and the knowledge that people with health limitations have more barriers to perform hygiene behavior [23], rather little is known about the differences between the general population and patients in, e.g., medical rehabilitation. What is known from before is the following: Patients with (pulmonary) comorbidities i.e. corona risk factors report significantly higher fear levels than those without comorbidities [20].” (lines 72ff).

…and the method section…

“Fears of a SARS-COV-2 Infection and COVID-19

Fears relating to the infection with SARS-COV-2 and possible consequences of COVID-19, such as fears of being infected, fears of getting seriously ill with COVID-19, and fears of infecting close ones, such as relatives, roommates or friends, were measured by three items. All three items were assessed on a five-point Likert scale from (1) “Never”, (2) “Rarely”, (3) “Sometimes”, (4) “Often”, and (5) “Always”. The items worded “How often to you fear of being infected with the coronavirus?”, How often do you fear of getting seriously ill with COVID-19” and “How often to you fear of infecting relatives/roommates or friends with COVID-19”. All three fear items were aggregated in terms of a means score which revealed good internal reliability with Cronbach's Alpha=.862.” (lines 233ff).

In general, the discussion appears to have more speculation, than interpretation through a theoretical lens, and it would be recommended that this, in particular, be revised.

RESPONSE BY THE AUTHORS: We appreciate this option and revised the entire discussion accordingly (lines 393ff). .

A thorough copy edit would be needed to strengthen the writing and grammar.

RESPONSE BY THE AUTHORS: Thanks, we copy edited the whole manuscript thoroughly.

Round 2

Reviewer 1 Report

The manuscript is quite improved, but I still recommend the following:

1.) Title:
OLD: Hygiene behavior...
NEW: Hygiene behaviors... OR SARS-CoV-2-preventive behaviors...
The reason for this: Is "avoiding large masses" a "hygiene" behavior? Also, the plurals give the hint that it's about multiple behaviors from the get-go. Finally, the "o" in "SARS-CoV-2" is written in lower case.

2.) Title:
OLD: ...in face...
NEW: ...in the face...

3.) Title:
OLD: ...corona...
NEW: ...COVID-19...

4.) Title:
OLD: Self-reported Compliance...
NEW: Self-reported compliance...

5.) Title: I strongly suggest the following changes:
OLD: ...Self-reported Compliance rates and associations with fear, SARS-COV-2 risk, and mental health
NEW: ...Self-reported compliance and associations with fear in a general population vs. a psychosomatic patients sample in Germany
Otherwise, readers might except predictions of COVID-19-risks and measures of mental health, which both the paper does not provide.

6.) Please format the sub-heading in the Introduction like those under Materials and Methods (e.g. "1.1 COVID-19-related fear and behavior" in italics).

7.) The authors switch to "adherence" here. Since we do not known whether the self-reported behaviors correspond to recommendations which have been AGREED upon between those who recommend and those who receive the recommendations (which would be needed to use the term "adherence"), I strongly recommend to stick to "compliance"/"comply" throughout the paper.

8.) Please add the non-random sampling procedures in both surveys as a major limitation of the study, including the fact that no survey response rates could be reported, and the associated probable critical selection biases.

Author Response

The manuscript is quite improved, but I still recommend the following:

RESPONSE BY THE AUTHORS: Thank you! We complied with all your suggestions and changed the manuscript accordingly (see attached track changes).

1.) Title:
OLD: Hygiene behavior...
NEW: Hygiene behaviors... OR SARS-CoV-2-preventive behaviors...
The reason for this: Is "avoiding large masses" a "hygiene" behavior? Also, the plurals give the hint that it's about multiple behaviors from the get-go. Finally, the "o" in "SARS-CoV-2" is written in lower case.

RESPONSE BY THE AUTHORS: Done.

2.) Title:
OLD: ...in face...
NEW: ...in the face...

RESPONSE BY THE AUTHORS: Done.

3.) Title:
OLD: ...corona...
NEW: ...COVID-19...

RESPONSE BY THE AUTHORS: Done.

4.) Title:
OLD: Self-reported Compliance...
NEW: Self-reported compliance...

RESPONSE BY THE AUTHORS: Done.

5.) Title: I strongly suggest the following changes:
OLD: ...Self-reported Compliance rates and associations with fear, SARS-COV-2 risk, and mental health
NEW: ...Self-reported compliance and associations with fear in a general population vs. a psychosomatic patients sample in Germany
Otherwise, readers might except predictions of COVID-19-risks and measures of mental health, which both the paper does not provide.

RESPONSE BY THE AUTHORS: Done.

6.) Please format the sub-heading in the Introduction like those under Materials and Methods (e.g. "1.1 COVID-19-related fear and behavior" in italics).

RESPONSE BY THE AUTHORS: Done.

7.) The authors switch to "adherence" here. Since we do not known whether the self-reported behaviors correspond to recommendations which have been AGREED upon between those who recommend and those who receive the recommendations (which would be needed to use the term "adherence"), I strongly recommend to stick to "compliance"/"comply" throughout the paper.

RESPONSE BY THE AUTHORS: Done.

8.) Please add the non-random sampling procedures in both surveys as a major limitation of the study, including the fact that no survey response rates could be reported, and the associated probable critical selection biases.

RESPONSE BY THE AUTHORS: Done.

Reviewer 2 Report

Thank you very much for presenting a clearer and easier-to-read version.

Author Response

Thank you very much for this positive feedback. English language and style received a minor spell check and we corrected the manuscript accordingly.